

# Leaf waxes in litter and topsoils along a European transect

Imke Schäfer[1], Verena Lanny[2], Jörg Franke[1], Timothy I. Eglinton[2], Michael Zech[3,4], Barbora Vysloužilová[5,6] and Roland Zech[1]

[1]Institute of Geography and Oeschger Centre for Climate Change Research, University of Bern, 3012 Bern, Switzerland
[2]Department of Earth Science, ETH Zurich, 8092 Zurich, Switzerland
[3]Landscape- & Geoecology, Faculty of Environmental Sciences, Technical University of Dresden, 01062 Dresden, Germany
[4]Institute of Agronomy and Nutritional Sciences, Soil Biogeochemistry, Martin-Luther University Halle-Wittenberg, 06120 Halle, Germany
[5]Institute of Archaeology of Academy of Science of the Czech Republic, Letenská 4, 11801 Praha 1, Czech Republic
[6]Laboratoire Image, Ville, Environnement, UMR 7362, CNRS/ Université de Strasbourg, 67083 Strasbourg Cedex, France

*Correspondence to*: Imke Schäfer (imke.schaefer@giub.unibe.ch)

**Abstract.** Lipid biomarkers are increasingly used to reconstruct past environmental and climate conditions. Leaf wax derived long chain *n*-alkanes and *n*-alkanoic acids may have great potential for reconstructing past changes in vegetation, but the factors that affect the leaf wax distribution in fresh plant material, as well as in soils and sediments are not yet fully
understood and need further research. We systematically investigated the influence of vegetation and soil depth on leaf waxes in litter and topsoils along a European transect. Our deciduous forest sites are often dominated by the *n*-$C_{27}$ alkane and *n*-$C_{28}$ alkanoic acid. Conifers produce few *n*-alkanes, but show high abundances of the $C_{24}$ *n*-alkanoic acid. Grasslands are characterized by relatively high amounts of $C_{31}$ and $C_{33}$ *n*-alkanes and $C_{32}$ and $C_{34}$ *n*-alkanoic acids. Chain length ratios thus may allow to distinguish between different vegetation types, but caution must be exercised given the large species-specific
variability of chain length patterns. An updated endmember model with a new *n*-alkane ratio is provided to illustrate, and tentatively account for, degradation effects on *n*-alkanes.

## 1 Introduction

To improve our understanding of ongoing environmental changes and to predict more precisely consequences of future climate change, it is important to investigate the magnitude of, and interactions between, climate and environmental
variations in the past. Lipid biomarkers are well preserved in many geological archives and are increasingly used for paleoclimate and paleoenvironmental reconstructions (Eglinton and Eglinton, 2008). Long chain *n*-alkanes (> $C_{25}$) and *n*-alkanoic acids (> $C_{20}$) for example are essential constituents of epicuticular leaf waxes and thus serve as specific biomarkers for higher terrestrial plants (Eglinton and Hamilton, 1967; Otto and Simpson, 2005).

Leaf wax *n*-alkanes typically show an odd-over-even predominance (OEP; Eglinton and Hamilton, 1967). The relative odd
homologue abundance may be useful to discriminate between different vegetation types: $C_{27}$ and $C_{29}$ have been reported to be predominant in leaf waxes of trees and shrubs, whereas $C_{31}$ and $C_{33}$ mostly derive from grasses and herbs (Maffei, 1996, Maffei et al., 2004; Rommerskirchen et al., 2006, Zech et al., 2009; Lei et al., 2010; Kirkels et al., 2013). Various *n*-alkane ratios have been proposed and used for paleovegetation reconstruction (Schwark et al., 2002; Zech et al., 2009; Lei et al., 2010; Schatz et al., 2011; Wiesenberg et al. 2015). Where pollen are preserved, leaf wax and pollen records are in good
agreement (e.g. Brincat et al., 2000; Schwark et al., 2002; Zech et al., 2010, Tarasov et al., 2013). In contrast, a recent study that summarises *n*-alkane patterns in modern plants from all over the world, showed no discrimination power for vegetation reconstruction at a global scale (Bush and McInerney, 2013), so regional calibration studies may be more appropriate. Additionally, the accuracy of *n*-alkane records remains somewhat uncertain due to several potential pitfalls:

i) Leaf wax production and concentration can vary widely between species (e.g. Diefendorf et al., 2011; Bush and
McInerney, 2013). Moreover, species abundance in an ecosystem controls leaf wax signals in soils and sediment.





ii) Several studies reported long chain *n*-alkanes not only in leaves, but also in other plant parts (roots, stems, blossoms; e.g. Wöstmann, 2006, Jansen et al., 2006, Kirkels et al., 2013, Gocke et al., 2013), but in these studies the patterns show preferential synthesis of shorter chains ($<n$-$C_{25}$) with low OEPs, and much lower *n*-alkane concentrations (three to ten times) than in leaves.

iii) Leaf waxes are affected by mineralization and degradation (e.g. Zech et al., 2009; Nguyen Tu et al., 2011; Zech et al., 2011a). As OEP values become lower during degradation, Zech et al. (2009) and Buggle et al. (2010) proposed procedures to quantify and correct *n*-alkane ratios for degradation using the OEP.

iv) Apart from vegetation type, many environmental parameters may influence leaf wax patterns, for example temperature and precipitation (e.g. Poynter et al., 1989; Sachse et al., 2006; Tipple and Pagani, 2013; Bush and McInerney, 2015), as

well as radiation, nutrient and water availability, salinity, mechanical stress and pollution (e.g. Shepherd and Wynne Griffiths, 2006; Guo et al., 2014). Sachse et al. (2006) and Duan and He, (2011) found longer chain lengths at lower latitudes, which could indicate (i) enhanced loss of shorter *n*-alkanes with increasing evaporation and (ii) preferential production of long chain *n*-alkanes, providing better protection against evaporation at higher temperatures and radiation. Schefuß et al. (2003) reported a higher *n*-$C_{31}$ vs. *n*-$C_{29}$ abundance in dust samples from drier regions along the West African

margin and suggested that humidity may be the driving factor. Bush and McInerney (2015) showed a correlation between ACL and temperature along a transect throughout the central United States and concluded that temperature is directly responsible for the synthesis of longer chain length.

So far, only few studies have investigated homologue long chain *n*-alkanoic acid patterns in plants, soils and sediments (e.g. Almendros et al., 1996; Marseille et al., 1999; Bull et al., 2000b; Zocatelli et al., 2012; Feakins et al., 2014, Wiesenberg et al.

2015). Leaf wax *n*-alkanoic acids have a distinct even-over-odd predominance (EOP; Eglinton and Hamilton, 1967). Zocatelli et al. (2012) found *n*-$C_{26}$ to be predominant in grassland soil relative to forest soil, whereas the forest soils contained *n*-$C_{22}$ to *n*-$C_{28}$ in greater amounts. Since *n*-alkane patterns alone seem to not always allow a reliable conclusion about the dominant vegetation type, there is an urgent need to more systematically investigate and understand the factors controlling the homologue *n*-alkanoic acid patterns.

For the present study, we collected and analysed litter samples in deciduous and coniferous forests and two topsoil layers below grasslands, as well as below deciduous and coniferous forests (0-3 cm depth, 3-10 cm depth), because one would expect the strongest changes in the original plant derived leaf wax patterns throughout degradation and microbial reworking of organic material in the topsoil. As the vegetation mixture along our transect never consists and probably in the past never consisted solely of one vegetation type we defined forests with a dominance (>80 %) of deciduous/ coniferous trees as

deciduous or coniferous forests, respectively. This represents the natural past vegetation composition and may allow estimating the main contributing vegetation type to the soil based on *n*-alkyl leaf wax biomarker distribution.

Specifically our study aims to evaluate

(i) the role of different vegetation types for *n*-alkane and *n*-alkanoic acid chain length patterns and

(ii) effects of degradation on leaf wax patterns.

## 2 Material and Methods

### 2.1 Geographical setting and sampling

In 2012 and 2014, we collected litter and topsoil samples (Ah1: 0-3 cm; Ah2: 3-10 cm) from 26 locations along a transect in Central Europe (Fig. 1). The samples BRO, HUB, HUG, HUM, HUR and KOC were kindly provided by B. Vysloužilová, more information about sampling and regional setting of these locations can be found in Vysloužilová et al. (2015). The

study area is in general characterized by relatively mild temperatures and moderate rainfall. The mean annual air temperature along the transect ranges from 5.5 to 11.0° C, and mean annual precipitation ranges from 470 to 1700 mm (see Table S1 for




individual data and data source). Altitudes range from 899 to 16 m above sea level. The natural vegetation consists of grasslands in the south of the transect and a higher amount of deciduous broadleaf and mixed forests (varying percentages of deciduous trees and conifers) in the north. Also, the proportion of evergreen conifers increases northwards.

We sampled soils in forests with a dominance of deciduous and coniferous trees, respectively, as well as soils below grasslands (referred to as "dec", "con" and "grass" in the following text and figures). Photographs of the sampling sites and descriptions of the dominant vegetation are provided in Table S2. For our dec and con sites, we could collect litter samples, but the grass sites virtually had no litter. Sampling sites were chosen in forests with large old trees, indicating a stable environment for more than approximately 30 years, limiting the risk that a vegetation change might have influenced the leaf wax signal in the soil, as leaf waxes are stable over long time periods (e.g. Derenne and Largeau, 2001). Please note that our

grass sites can be dominated by grasses, herbs or heaths. Soil and litter samples at each site were composites from three sampling points approximately five to seven meters from each other.

### 2.2 Lipid analysis

Lipids were extracted from 1-6 g freeze-dried and grinded samples by microwave extraction with 15 ml dichloromethane (DCM)/methanol (MeOH) (9:1) at 100 °C for 1 h. Each total lipid extract was passed over a pipette column filled with

15 aminopropyl silica gel as stationary phase (Supelco, 45 µm). The apolar fraction (incl. $n$-alkanes) was eluted with hexane, more polar compounds (e.g. alcohols) with DCM/MeOH (1:1), and acids (incl. $n$-alkanoic acids) with 5% acetic acid in diethyl ether. The $n$-alkanes were purified by passing the apolar fraction over a pipette column filled with activated AgNO$_3$ impregnated silica gel (to retain unsaturated compounds) and another pipette column filled with zeolite (Geokleen). After drying, the zeolite (containing straight chain compounds) was dissolved in HF and the $n$-alkanes were recovered by liquid-

20 liquid extraction with hexane. For quantification, the $n$-alkane fractions were spiked with a known amount of the 5α-androstane and analysed with an Agilent 7890 gas chromatograph (GC) equipped with a VF1 column (30 m length × 0.25 mm i.d., 0.25 µm film thickness) and a flame ionization detector (FID).

The $n$-alkanoic acids were converted to fatty acid methyl esters (FAMEs) with MeOH/HCl (95/5; 70°C, overnight). The FAMEs were recovered by liquid-liquid extraction with hexane and cleaned over silica, AgNO$_3$ and zeolite columns as

described above before quantification with GC-FID. Unfortunately, due to some problems during FAME preparation, not all samples were available for methylation (missing samples: BRO, HUB, HUG, HUM, HUR, KOC).

### 2.3 Leaf wax proxies

Total $n$-alkane and $n$-alkanoic acid concentrations ($c_{tot}$) are calculated as sum of C$_{25}$ to C$_{35}$ and C$_{20}$ to C$_{34}$, respectively and given in µg/g dry weight (dw).

Changes in the average chain length (ACL) of $n$-alkyl lipids can show changes in the input of vegetation type. The ACL was determined by modifying the equation of Poynter et al. (1989). We used odd chain lengths only for $n$-alkanes (Eq. 1), and even chain lengths for $n$-alkanoic acids (Eq. 2), respectively.

$$ACL\,(n-alkanes) = \frac{27 \times n-C_{27} + 29 \times n-C_{29} + 31 \times n-C_{31} + 33 \times n-C_{33}}{n-C_{27} + n-C_{29} + n-C_{31} + n-C_{33}} \tag{1}$$

$$ACL\,(n-alkanoic\,acids) = \frac{24 \times n-C_{24} + 26 \times n-C_{26} + 28 \times n-C_{28} + 30 \times n-C_{30} + 32 \times n-C_{32}}{n-C_{24} + n-C_{26} + n-C_{28} + n-C_{30} + n-C_{32}} \tag{2}$$

The odd-over-even predominance (OEP) of the $n$-alkanes (Eq. 3) and the even-over-odd predominance (EOP) of the $n$-alkanoic acids (Eq. 4) can be used as proxy for degradation and was determined after Hoefs et al. (2002):

$$OEP = \frac{n-C_{27} + n-C_{29} + n-C_{31} + n-C_{33}}{n-C_{26} + n-C_{28} + n-C_{30} + n-C_{32}} \tag{3}$$

$$EOP = \frac{n-C_{24} + n-C_{26} + n-C_{28} + n-C_{30} + n-C_{32}}{n-C_{23} + n-C_{25} + n-C_{27} + n-C_{29} + n-C_{31}} \tag{4}$$





High OEP values are characteristic for fresh plant material, with ongoing SOM degradation in the topsoil the OEP values decrease (e.g. Buggle et al. 2010; Zech et al. 2009, 2011b).

## 2.4 Statistical analysis

We conducted an Analysis of Variance (ANOVA) to check for significant differences ($\alpha=0.05$) between depths horizons
within the same vegetation type and between vegetation types within the same horizon, respectively. If the ANOVA indicated significant differences in the means, we applied a so-called "post-hoc" test to identify which of the means differ, accounting for the effect of multiple testing. The appropriate test was selected as recommended by Field, 2013: For samples with equal size and equal variance, we applied the Tukey's Honest Significance test (Tukey, 1949). In case of equal variance and unequal sample size, we used the Hochberg test (Hochberg, 1988) and for unequal variances the Games-Howell test
(Games and Howell, 1976). Equal variances were tested with a Levene Test (Levene, 1960).

## 3 Results

### 3.1 Leaf wax *n*-alkane abundances and chain length patterns

All samples show a dominance of long ($>C_{25}$) odd chain *n*-alkanes, characteristic for epicuticular leaf waxes (e.g. Eglinton and Hamilton, 1967; Rieley et al., 1991; Collister et al., 1994). Total *n*-alkane concentrations ($C_{tot}$) range from 0.4 to 1468
15  µg/g dw (Table S3). Such concentrations and huge variability are in agreement with published data from fresh plant material (e.g. Diefendorf et al., 2011; Hoffmann et al., 2013) and from soil and sediments (e.g. Marseille et al., 1999; Freeman and Colarusso, 2001; Liebezeit and Wöstmann, 2009). Differences exist depending on the vegetation type and litter/soil horizon (Fig. 2a), but are mostly not significant (Table S4).

Differences in chain length patterns between deciduous forests, coniferous forests and grasslands are illustrated in Fig. 3 for
litter (a), Ah1 (b) and Ah2 (c). The deciduous forest samples are strongly dominated by *n*-$C_{27}$, although its relative abundance decreases from litter to Ah2. However, most of our sampling sites are dominated by beech trees (L11, L13, L14, L16, L17, L18, L20 and L23) that are known to produce mostly *n*-$C_{27}$ (Bush and McInerney, 2013 and references therein). To check whether the observed *n*-$C_{27}$ dominance could be explained only with the presence of beech we performed the same correlations and plots as mentioned above, excluding the beech dominated sites. Figure S5 shows that the dominance of *n*-
$C_{27}$ is here less pronounced and get lost in Ah2. Grass sites are dominated by *n*-$C_{31}$ and are also characterized by high abundances of *n*-$C_{33}$ compared to deciduous sites. Our con sites show a pattern very similar to that of our grass sites.

The ACLs of the grass and con sites are significantly higher than those of the dec sites (Fig 4a; grass: Ah1= 30.5; Ah2= 30.3; dec: Ah1= 28.6; Ah2= 29.6, see Table S6 for p-values). Without the beech dominated sites, the dec sites ACL shifts to higher values, but they are still significantly lower than mean ACLs of the grass sites (Figure S 7a). However, significant
differences get lost in Ah2 (Table S6). Our dec litter samples have lower ACLs (27.5) than the Ah1 (28.6) and Ah2 (29.6). Our grass samples show almost no decrease in ACL from Ah1 (30.5) to Ah2 (30.3), the con sites likewise show no significant decrease from L to Ah2 (Table S6).

To study past changes in dec vs. grass vegetation, various *n*-alkane ratios have been proposed and used (e.g. Zhang et al., 2006; Lei et al., 2010; Bush and McInerney, 2013; Zech et al., 2013a; Zech et al., 2013b). We tested several *n*-alkane ratios
(*n*-$C_{33}$/(*n*-$C_{27}$+*n*-$C_{33}$), (*n*-$C_{31}$+*n*-$C_{33}$)/(*n*-$C_{27}$+*n*-$C_{31}$+*n*-$C_{33}$), (*n*-$C_{31}$+*n*-$C_{33}$)/(*n*-$C_{27}$+*n*-$C_{29}$+*n*-$C_{31}$+*n*-$C_{33}$)) and found the largest differences between our grass and dec samples for the ratio (*n*-$C_{31}$+*n*-$C_{33}$)/(*n*-$C_{27}$+*n*-$C_{31}$+*n*-$C_{33}$). It is low in the dec samples and high in the grass samples (Fig. 4b, dec L: 0.08; Ah1: 0.29; Ah2: 0.56; grass Ah1: 0.79; Ah2: 0.78). Differences between dec and grass are significant in Ah1 and Ah2 (Table S8). Without the beech dominated sites the ratio becomes higher for dec, but still shows significant differences between dec and grass (Figure S7b, Table S8).



The OEP (or CPI, carbon preference index, which is very similar to the OEP) is often regarded as proxy for the preservation status of the leaf wax derived $n$-alkanes (e.g. Huang et al., 1996; Tipple and Pagani, 2010; Vogts et al., 2012; Wang et al., 2014 and references therein). High OEPs are characteristic for fresh plant material and modern soils (Kirkels et al., 2013; Diefendorf et al., 2011; Collister et al., 1994), whereas low OEPs indicate degradation of $n$-alkanes during pedogenesis and

5 early diagenesis (Marseille et al., 1999; Freeman and Colarusso, 2001; Buggle et al., 2010; Zech et al., 2011a; Wang et al., 2014). OEPs in our samples range from 3 to 32.8, typical for fresh plant material and soils (Table S3). Values significantly decrease from litter (18.4) to Ah1 (12.1) and Ah2 (6.8) for our dec sites, a minor decrease can be observed in our grass sites (Fig. 4c, Table S9). Significant differences occur between dec and grass sites in all horizons (Table S9).

### 3.2 Leaf wax $n$-alkanoic acid abundances and chain length patterns

All samples show high abundances of long ($>C_{20}$) even chain $n$-alkanoic acids (Table S10), characteristic for epicuticular leaf waxes (Eglinton and Hamilton, 1967). Many samples also have large amounts of $C_{16}$ and $C_{18}$, yet those are ubiquitous and cannot be considered as leaf wax biomarkers. Total $n$-alkanoic acid concentrations ($c_{tot}$, refers here to the sum of $C_{20}$ to $C_{34}$) range from 3 to 854 µg/g dw, consistent with previous studies (e.g. Marseille et al., 1999; Jandl et al., 2002). As for the $n$-alkanes, total $n$-alkanoic acid concentrations vary between the vegetation types and horizons (Fig. 2b). In general, $c_{tot}$

decreases from litter to Ah1 and Ah2. In contrast to the $n$-alkanes, highest $n$-alkanoic acid concentrations occur in our con samples. Decrease from Litter to Ah1 and Ah2 are only significant in our dec samples (Table S11).

The $n$-alkanoic acid chain length patterns show differences between the different vegetation types (Fig. 3d: litter; 3e: Ah1; 3f: Ah2). While our dec samples are dominated by $n$-$C_{28}$, the con samples show a maximum for the shorter homologue $n$-$C_{24}$ in the litter, Ah1 and Ah2. The grass sites have high abundances of $n$-$C_{24}$ to $n$-$C_{30}$, but when compared to dec and con, the

20 relative $n$-$C_{32}$ and $n$-$C_{34}$ abundances are distinct.

Significant differences in the ACL between the three vegetation types exist and reflect the above mentioned predominance of various homologues (Fig. 5a, Table S12). While our grass sites show a tendency to higher ACLs, the con sites tend to have the lowest values. Differences stay significant, even in Ah1 and Ah2 (Table S12).

On the basis of the above mentioned significant differences between the ACL and the three vegetation types we propose the

25 following three indices, dubbed CDG indices (Eq. 5,6,7) for coniferous forests (C), deciduous forests (D) and grasslands (G):

$$\text{Index C} = \frac{n-C_{24}}{n-C_{24} + n-C_{28} + n-C_{32} + n-C_{34}} \tag{5}$$

$$\text{Index D} = \frac{n-C_{28}}{n-C_{24} + n-C_{28} + n-C_{32} + n-C_{34}} \tag{6}$$

$$\text{Index G} = \frac{n-C_{32} + n-C_{34}}{n-C_{24} + n-C_{28} + n-C_{32} + n-C_{34}} \tag{7}$$

Index C ranges from 0.25 to 0.91 with the highest values for the con sites (Fig. 6a) and shows significant differences

between our con and dec samples, as well as between the con and grass in all horizons, but not between our dec and grass locations (Table S13). Index D ranges from 0.03 to 0.62 and has highest values for the dec sites (Fig. 6b). It differs significantly between all vegetation types in L, Ah1 and Ah2, except for con and grass sites in Ah2. It also significantly decreases from L to Ah1 and Ah2 in our dec sites (Table S14). Index G ranges from 0.00 to 0.37 and discriminates between forest and grass sites, with systematically higher values for the grass sites (Fig. 6c). Like Index D, Index G shows significant

differences between all three vegetation types in all horizons, except for the con and dec locations in Ah1. The Index likewise shows a significant decrease from L to Ah2 in our dec samples (Table S15). The CDG indices can conveniently be plotted in ternary diagrams, which illustrate the clusters for the different vegetation types and the scatter within the cluster (Fig. 7).

EOPs in our samples range from 1.2 to 9.9 (Table S16), typical for $n$-alkanoic acids that originate from epicuticular leaf

waxes and are found in soils (Killops and Killops, 2005). The EOP decreases significantly from litter to Ah1 and Ah2 in the





dec samples (L: 4.3; Ah1: 3.46; Ah2: 2.85 Fig. 5b; Table S16), and without significance in the con samples (L: 4.0; Ah1: 3.8; Ah2: 2.81).

## 4 Discussion

### 4.1 *n*-Alkane pattern in vegetation and topsoil

The lower $c_{tot}$ values of our con litter samples compared to the dec litter (median con: 30.9 µg/g dw; dec: 80.4 µg/g dw) are in good agreement with findings of much lower *n*-alkane abundances in conifer needles than deciduous leaves (e.g. Sachse et al., 2006; Zech et al., 2009; Diefendorf et al., 2011; Norris et al. 2013; Tarasov et al., 2013) and in forest soils below conifers (e.g. Almendros et al., 1996). Given these low reported *n*-alkane concentrations in conifer needles, we refer the *n*-alkane patterns in our con litter and topsoil samples to the *n*-alkane input from the understory, and we focus in the following

discussion on the differences between grass and dec sites. Our grass soils have very low *n*-alkane abundances (median Ah1, 3.5 µg/g dw), which we interpret to be an artefact of (former) plowing and admixture with inorganic soil material. Based on our data we therefore cannot infer a low *n*-alkane production in grass sites.

#### 4.1.1 *n*-Alkanes to distinguish between vegetation types

   The domination of *n*-$C_{27}$ in our dec samples in all horizons, and the relatively high concentration of *n*-$C_{31}$ and *n*-$C_{33}$ in our

grass samples, implies that the established source specific compounds, at least along our transect, allow conclusion on the vegetation type that generated them. However, Figure S5 proves that the pattern is less specific when the beech dominated sites are excluded. This supports former results and shows that *n*-$C_{27}$ is strongly produced by beech trees (Bush and McInerney, 2013 and references therein). The ACL and our proposed *n*-alkane ratio (*n*-$C_{31}$+*n*-$C_{33}$)/(*n*-$C_{27}$+*n*-$C_{31}$+*n*-$C_{33}$), show significant differences between our dec and our grass sites in Ah1 and Ah2, even when the beech dominated sites are

excluded (Table S6, S8). Although *n*-$C_{27}$ is not the dominant long odd chain *n*-alkane in Ah2 at the dec locations that are not dominated by beeches, its percentage in the dec samples is still higher than at the grass sites (Figure S5), whereas the percentage of *n*-$C_{31}$ and *n*-$C_{33}$ is the highest in the grass sites in Ah1 as well as in Ah2. So our results corroborate, at least for our studied transect, that the ACL and our proposed *n*-alkane ratio (*n*-$C_{31}$+*n*-$C_{33}$)/(*n*-$C_{27}$+*n*-$C_{31}$+*n*-$C_{33}$) allow to differentiate between the input of dec and grass vegetation. Care has to be taken when interpreting paleovegetation changes solely on the

dominance of one *n*-alkane compound over the others (e.g. proxies like $C_{max}$, Wiesenberg et al. 2015), because this might lead to an underestimation of the deciduous tree input, at least when beech trees were not the main contributors to the soil. Nevertheless, we strongly emphasize that our observed patterns are very likely a regional phenomenon and our results should not be transmitted to other regions with different climate and vegetation types, because *n*-alkane patterns do not work on a global scale (Bush and McInerney, 2013). Thus, our results underline the need of regional calibrations for the *n*-alkane

pattern, because they corroborate it`s potential for paleovegetation reconstruction on a regional base.

   Although significant differences occur between the dec and grass sites OEP in Ah1 and Ah2, we would not recommend to use the OEP as a proxy to distinguish between the two vegetation types, because it can very likely show significant decreases with ongoing soil depth, as it does in our dec samples (Table S9) so it is probably strongly influenced by degradation and microbial reworking.

#### 4.1.2 Influence of soil depth on the *n*-alkane pattern

   Although we observed no significant decreases in $C_{tot}$ from L to Ah1 and Ah2 for the dec and con sites and from Ah1 to Ah2 for all three vegetation forms (Table S4), the slightly decreasing trends in Figure 2a are most likely due to lipid degradation and admixture with inorganic soil material.





As stated above, the significant decrease in OEP from L to Ah2 in our dec samples (Fig. 4c, Table S9) is probably due to degradation effects. So, despite the wide range of OEPs in modern plants (Bush and McInerney, 2013 and references therein), the OEP can serve as a degradation proxy along our transect. The OEPs in our grass samples show no significant decrease from Ah1 (7.2) to Ah2 (6.5), but the degradation effects here are probably biased by plowing and mixing of Ah1

and Ah2.

The decrease in the relative percentage of the dominant *n*-alkane(s) in our dec and grass samples from L to Ah1 and from Ah1 to Ah2 (Fig. 3 a-c) as well as the significant increase in ACL from L to Ah2 in the dec samples (Fig. 4a, Table S6) is probably another indication for the effect of degradation on the *n*-alkane pattern. Our grass samples show an insignificant, but decreasing trend in ACL from Ah1 (30.5) to Ah2 (30.3). These observed changes are consistent with the notion that the

more abundant homologues are preferentially degraded and lost during pedogenesis. This affects *n*-alkane patterns in soils and sediments (Zech et al., 2009, Zech et al., 2013a; Zech et al., 2013b). Degradation also affects our ratio ($n$-$C_{31}$+$n$-$C_{33}$)/($n$-$C_{27}$+$n$-$C_{31}$+$n$-$C_{33}$) which is expressed in the significant increases in the dec samples from litter (0.08) to Ah1 (0.3) to Ah2 (0.56) and in the slightly decreasing trend in the grass samples from 0.84 to 0.80 (Fig 4b, Table S8).

In order to illustrate and correct for degradation effects, Zech et al. (2009) proposed an endmember model, which was later

modified by Zech et al. (2013a) and Zech et al. (2013b). We added our dec and grass samples to the dataset for Europe, provided by Zech et al. (2013b). Figure 8 shows the new endmember plot and illustrates that our *n*-alkane ratio differs between grass and dec samples and that it changes depending on the OEP, i.e. with degradation. As described already above, the *n*-alkane ratio is wider for grass samples and lower values are more typical for dec. With increasing degradation, differences seem to become less: the trend lines, or "degradation lines", for grass and dec converge. Again, we tested

whether the endmember plot can be explained only by the presence of beech by applying the same model and plot without the beech dominated sites. The dec degradation line shifts upward, closer to the grass degradation line (Figure S17), and $R^2$ drops from 0.25 to only 0.1, but it still shows a separation of the dec samples from the grass samples. Four sites plot particularly high above the dec degradation line and deserve a closer look. Sample location L19 is a birch forest surrounded by fields, and L26-2 dec is an open forest with birch and oak trees, with a lot of grasses in the understory. For both sites, the

*n*-alkane ratios for Ah1 and Ah2 plot much closer to the grass degradation line than the litter samples, and we speculate that both sites may have been grasslands in the past that were only recently reforested. Since turnover times of *n*-alkanes are in the order of decades (e.g. Amelung et al. 2008 and references therein, Wiesenberg et al. 2004) we would expect to see the *n*-alkane pattern prior to reforestation in the upper soil. Unfortunately we do not have information about former land use at our study locations to verify this speculation. The sample locations L22 (acer, elder, ash, poplar) and L26-1 (acer, oak, beech,

fir) are characterized by litter samples that plot close to the grass degradation line. We cannot exclude that these litter samples and sites are affected by *n*-alkane input from grasses, but likely the data simply reflect the large species-specific variability of *n*-alkane patterns reported repeatedly in the literature (e.g. Diefendorf et al. 2011; Bush and McInerney 2013).

In summary, our results show that

(i) *n*-alkane patterns are systematically different between the investigated dec and grass sites,

(ii) soil depth/degradation affects the homologue patterns, and

(iii) endmember modelling is a useful tool for paleovegetation reconstruction along our transect, but one needs to be aware of the uncertainties related mainly to the large species-specific variability of the *n*-alkane patterns. However, the fact that coniferous trees produce only few *n*-alkanes makes respective paleovegetation reconstructions "blind" for coniferous trees.

## 4.2 *n*-Alkanoic acid pattern in vegetation and topsoil

To the best of our knowledge, this is the first study which systematically investigates long chain *n*-alkanoic acid patterns in litter and topsoil along a transect that encompasses a range of environmental conditions and vegetation types. Since differences in con concentration compared to dec are much more pronounced in the topsoil than in the litter, we infer that





better preservation of *n*-alkanoic acids in soils under coniferous forests is the reason for the observed differences, and not higher alkanoic acid production by conifers. This is further consistent with studies showing better preservation of alkanoic acids in soils with low pH typical for coniferous forests, while *n*-alkanes are better preserved in soils with a high pH, more typical for deciduous forests (Bull et al., 2000a; Zocatelli et al., 2012). We again attribute the low $c_{tot}$ in our grass sites

mainly to plowing and admixture with inorganic soil material.

### 4.2.1 *n*-Alkanoic acids to distinguish between vegetation types

The *n*-alkanoic acid distribution in the vegetation types implies that specific compounds can be used to characterize them (Fig. 3d, e, f). The *n*-$C_{24}$ alkanoic acid can represent the input of conifers in L, Ah1 and Ah2, *n*-$C_{28}$ shows the contribution of deciduous trees in all horizons and the relative amount of *n*-$C_{32}$ and *n*-$C_{34}$ can be used to estimate the grass contribution.

From that, we suggest the CDG Indices. They show strong differences between the three vegetation types (Fig. 6, Table S13-15) which are significant in nearly all horizons, apart from Index C which does not allow a distinction between dec and grass sites (Table S13). The ternary plots of the three indices visualize the discrimination potential by showing clusters for the different vegetation types, although we have to emphasize that outliers exist (Fig. 7 a-c). Index C based on the dominance of the shorter chain *n*-alkanoic acid $C_{24}$, which might be stronger affected by microbial degradation and reworking compared to

the longer chain counterparts *n*-$C_{28}$, *n*-$C_{32}$ and *n*-$C_{34}$ that are included in the D and G Index (section 4.2.2). The ACL of the *n*-alkanoic acids on the other hand may not be a particularly useful proxy for paleovegetation, because the observed differences are small (although they are significant, Table S12) and mixing a con and grass signal could falsely yield a dec signal.

### 4.2.2 Influence of soil depth on the *n*-alkanoic acid pattern

The significant decrease in $C_{tot}$ of our dec and con samples from L to Ah2 (Table S11) is most likely attributed to enhanced degradation effects on the acids with ongoing soil depth.

The preferential loss of *n*-$C_{28}$ (*n*-$C_{24}$) in our dec samples (con samples) can be visualized by comparing the homologue patterns of litter, Ah1 and Ah2 (Fig. 3d, e, f). This degradation effect is not documented in a significant change of the ACL, only the con sites show a slight, but not significant increasing trend with ongoing soil depth (Fig. 5a, Table S12). The

degradation effect of a certain homologue on the CDG indices is illustrated in Fig. 6 and 7. Index D, for example, is high for our dec litter samples, but significantly decreases from litter to Ah1 and Ah2 (Table S14). The same applies for Index C and G with regard to the con and grass samples, respectively, although the changes are not significant. With the preferential loss of the most abundant compound (*n*-$C_{28}$ for dec and *n*-$C_{24}$ for con) the respective characteristic index decreases, but unavoidably the other two indices increase. This is illustrated by the clusters moving closer together (Fig. 7b-c, d).

Nevertheless, they still allow to discriminate between the three vegetation types as at least Index D and G show significant differences between all three vegetation types in all horizons (Table S14, S15). All three indices show a significant decrease (Index D) or increase (Index C and Index G) with soil depth in our dec samples which implies that they are more prone to degradation under the more alkaline deciduous forest soils. The grass samples do not show changes in ACL or in the indices from Ah1 to Ah2, which we again refer to plowing.

The significant decrease in EOP from litter to Ah1 and Ah2 in our dec samples (dec litter: 4.3; Ah1: 3.46; Ah2: 2.85; Fig. 5b) resembles the decrease of the OEP for the *n*-alkanes and suggests that the most abundant (even-numbered) compounds are preferentially degraded during pedogenesis. The decrease in EOP in the con samples is not significant, which we again refer to a better preservation of *n*-alkanoic acids in the acidic soils under conifers. Nevertheless, Figure 5b indicates a decreasing trend in the EOP in our con samples from L to Ah2, probably due to slight degradation effects on the acids in the

con Ah1 and Ah2 samples. Grass samples do not show a trend in EOP, which is most likely because of the plowing that affected these sites. Like the OEP, the EOP might thus serve as proxy for degradation.




Although our results demonstrate that the leaf wax derived *n*-alkanoic acids in soils under coniferous forests are less prone to degradation compared to soils under deciduous forests, the risk still exists, that the leaf wax contribution from coniferous trees to soil and sedimentary archives might be underestimated, when the alkanoic acid pattern is not corrected for degradation. The same applies for the deciduous forests and probably also for the grass sites.

Overall, our results show that

(i) *n*-alkanoic acid patterns are significantly different between the investigated dec, con and grass sites,

(ii) the specific CDG indices might be valuable proxies for paleovegetation, and

(iii) degradation affects the homologue patterns and CDG indices, at least in our dec samples, so that procedures to correct for degradation need to be developed and tested.

**5 Conclusions**

We have systematically investigated leaf wax derived long chain *n*-alkane and *n*-alkanoic acid patterns in litter and top soils along a European transect. We found out that:

1. Both compound classes show distinct differences depending on the type of vegetation. The vegetation signal is not only found in the litter, but can also be preserved to some degree in the topsoil. Our grass sites contain more $n$-C$_{31}$ and $n$-C$_{33}$

alkanes than our dec sites, but less $n$-C$_{27}$. The ratio ($n$-C$_{31}$+$n$-C$_{33}$)/($n$-C$_{27}$+$n$-C$_{31}$+$n$-C$_{33}$) seems to be most suitable to distinguish between those two vegetation types in our study area. Litter and soil samples in coniferous forests are probably biased by the understory, so vegetation reconstructions solely based on the *n*-alkane pattern are blind for coniferous trees. Nevertheless, the *n*-alkanes show a great potential for paleovegetation reconstruction along our transect, but the species-specific absolute and relative variability of the homologue abundances need to be taken into account.

We propose three *n*-alkanoic acid indices to distinguish and possibly quantify contributions from the three investigated vegetation types: Index C is the relative abundance of the C$_{24}$ *n*-alkanoic acid and represents the input of coniferous trees. Index D is the relative abundance of the C$_{28}$ *n*-alkanoic acid and is particularly high in litter and in topsoil of deciduous forests. The relative abundance of the C$_{32}$ and C$_{34}$ *n*-alkanoic acids is expressed as Index G and quantifies the contribution from grasses and herbs.

2. The homologue patterns of leaf waxes change from litter to Ah1 and Ah2. Although we cannot completely rule out effects related to possible land use and vegetation change in the past, the overall consistent trends imply that degradation plays an important role. Degradation cannot only lower the OEP and EOP of *n*-alkanes and *n*-alkanoic acids, respectively, it also reduces the vegetation specific differences of the homologue patterns. An updated endmember model is suggested to account for degradation effects on *n*-alkanes, but similar procedures yet need to be developed and tested for the *n*-alkanoic acids

before their potential for paleovegetation reconstructions can be fully exploited.

Overall, our findings suggest that combined investigations of *n*-alkane and *n*-alkanoic acid distributions on a regional scale have great potential for paleovegetation reconstruction, although degradation effects need to be taken into account. Particularly, with regard to the *n*-alkanoic acids, more research is needed to a better understanding of those effects.

*Acknowledgements.* We thank P. Neitzel, who contributed in large part to the work in the field and in the laboratory at ETH
Zurich, and Q. Lejeune for support in the field, as well as C. Magill for scientific discussions. C. Diebold helped with the laboratory work at the University of Bern. We also acknowledge L. Wüthrich and M. Bliedtner for helpful discussions. The research was funded by the Swiss National Science Foundation (PP00P2 150590).



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





**Figure 1:** Sample locations (black dots) along the transect (map source: US National Park Service; Esri, HERE, DeLorme, MapmyIndia, OpenStreetMap contributors and the GIS user community).




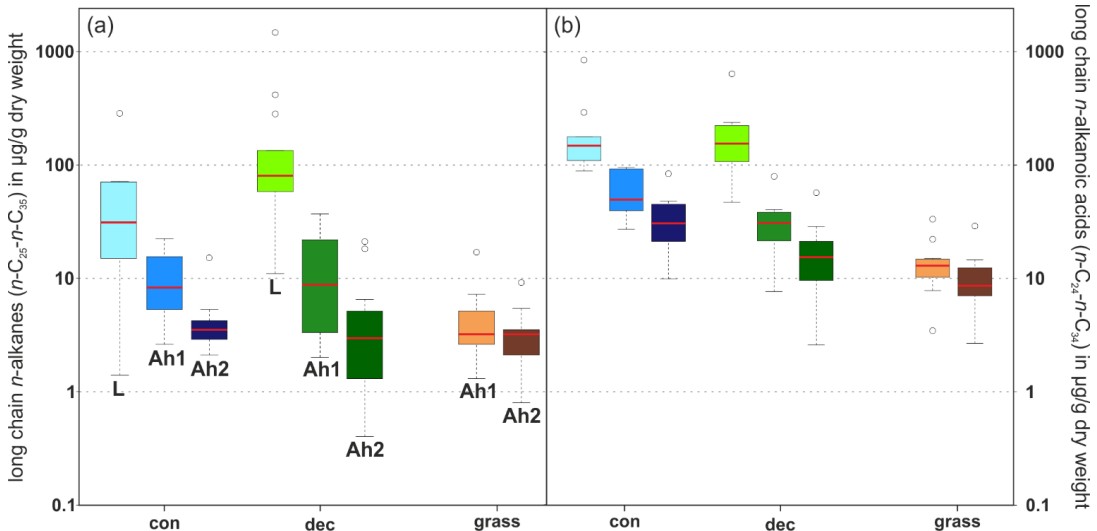

**Figure 2:** Total concentrations of (a) *n*-alkanes and (b) *n*-alkanoic acids in µg/g dry weight; con: coniferous forest sites (n=9); dec: deciduous forest sites (n=14); grass: grassland sites (n=22); L=litter; Ah1=topsoil 1 (0-3 cm); Ah2=topsoil 2 (3-10 cm). Box plots show median (red line), interquartile range (IQR) with upper (75%) and lower (25%) quartiles, lowest datum still within 1.5 x IQR of lower quartile, highest datum still within 1.5 x IQR of upper quartile.

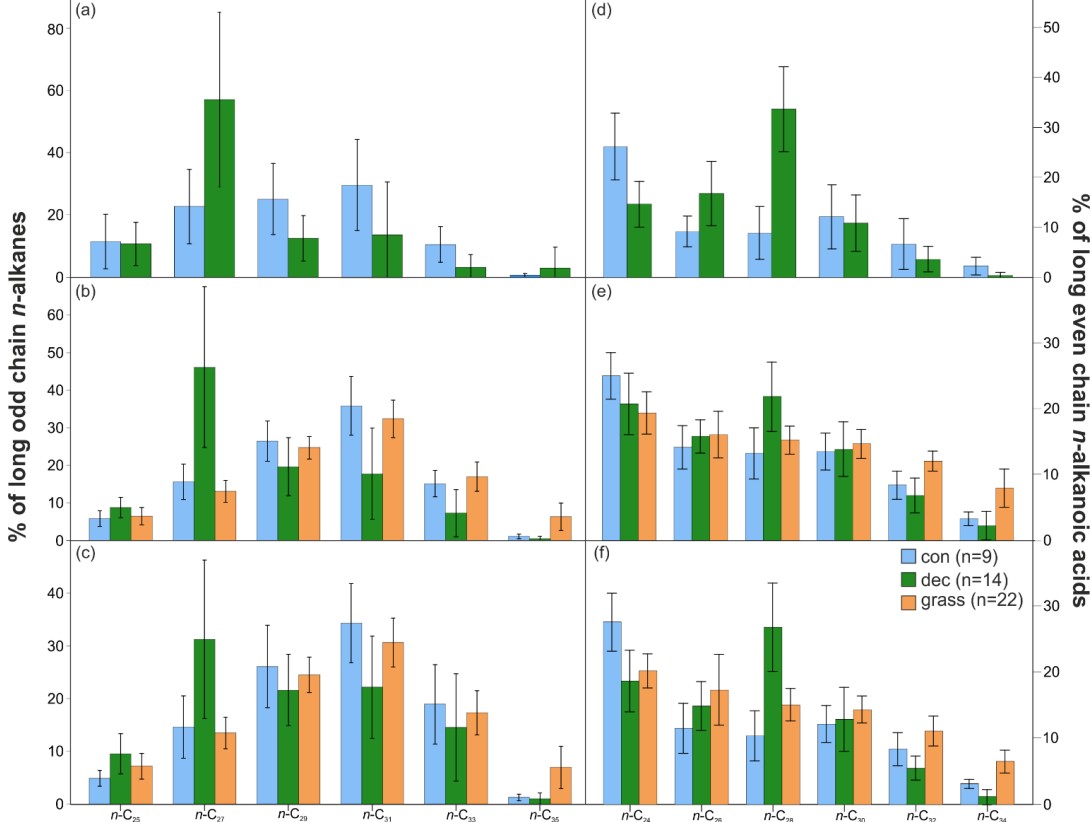

**Figure 3:** Chain length patterns for odd long chain *n*-alkanes in (a) litter, (b) Ah1 and (c) Ah2, and long and even chain *n*-alkanoic acids in (d) litter, (e) Ah1, (f) Ah2.





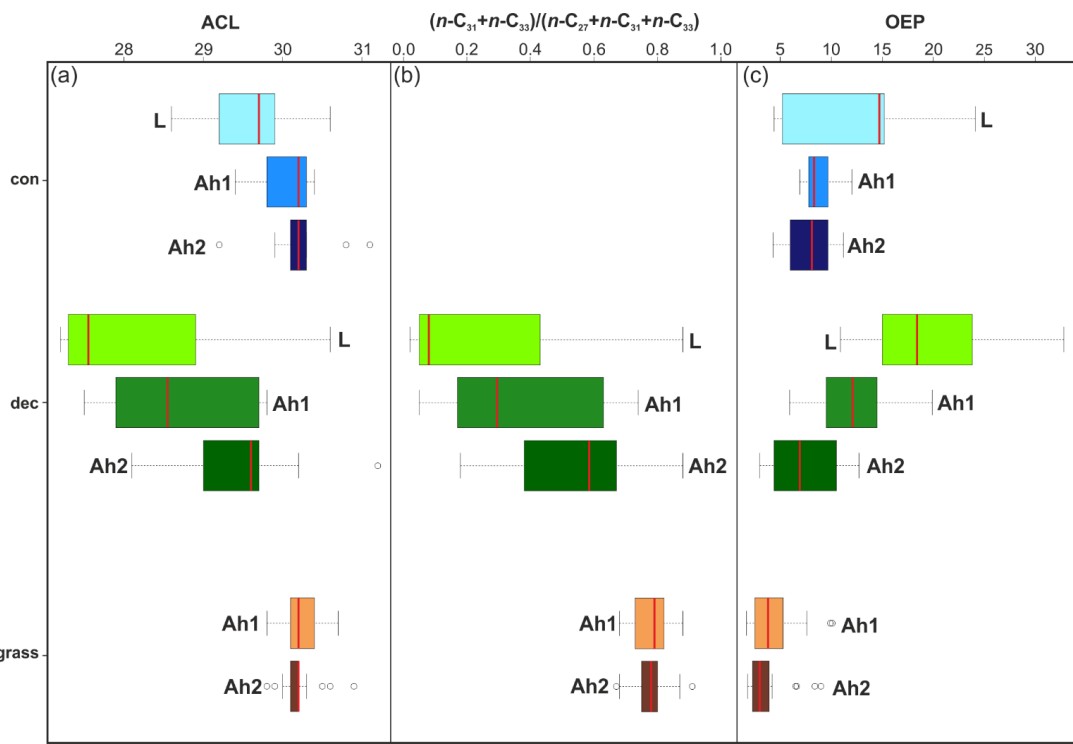

**Figure 4:** Box plots (a) *n*-alkane ACL, (b) ratio $(n\text{-}C_{31}+n\text{-}C_{33})/(n\text{-}C_{27}+n\text{-}C_{31}+n\text{-}C_{33})$, and (c) OEP. Con: coniferous forest sites (n=9); dec: deciduous forest sites (n=14); grass: grassland sites (n=22); L=litter; Ah1=topsoil 1 (0-3 cm); Ah2=topsoil 2 (3-10 cm). Box plots show median (red line), interquartile range (IQR) with upper (75%) and lower (25%) quartiles, lowest datum still within 1.5 x IQR of lower quartile, highest datum still within 1.5 x IQR of upper quartile.

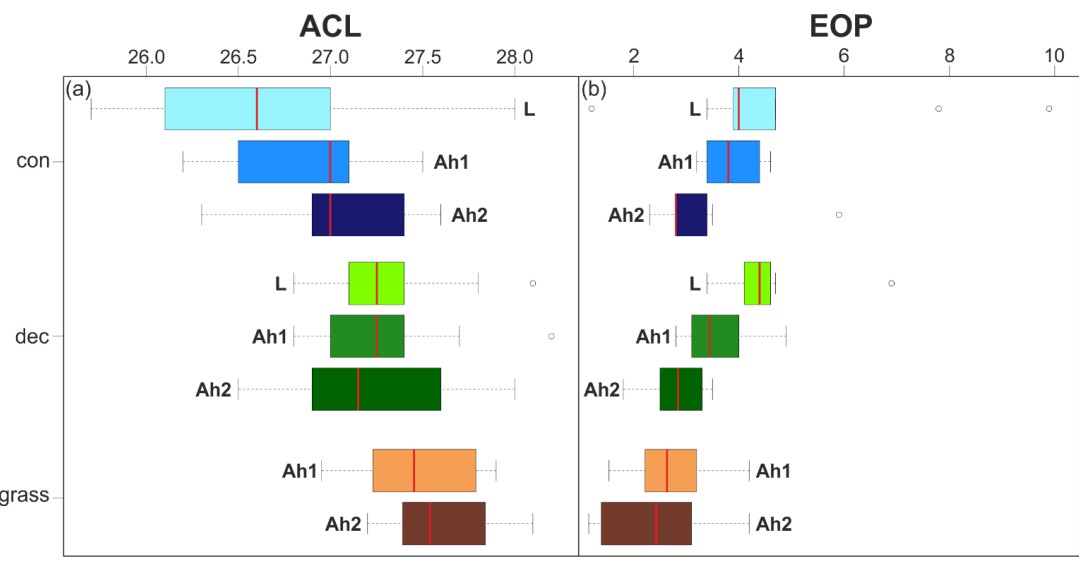

**Figure 5:** Box plots for (a) *n*-alkanoic acid ACL and (b) EOP. Con: coniferous forest sites (n=9); dec: deciduous forest sites (n=14); grass: grassland sites (n=14); L=litter; Ah1=topsoil 1 (0-3 cm); Ah2=topsoil 2 (3-10 cm). Box plots show median (red line), interquartile range (IQR) with upper (75%) and lower (25%) quartiles, lowest datum still within 1.5 x IQR of lower quartile, highest datum still within 1.5 x IQR of upper quartile.



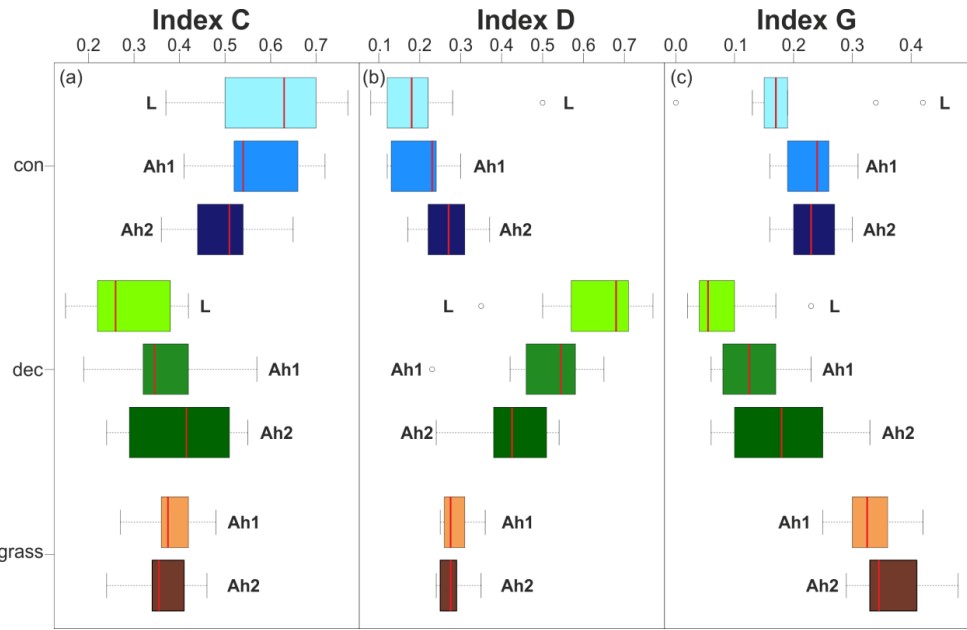

**Figure 6: Box plots for (a) Indices C, (b) D and (c) G. Con: coniferous forest sites (n=9); dec: deciduous forest sites (n=14); grass: grassland sites (n=14); L=litter; Ah1=topsoil 1 (0-3 cm); Ah2=topsoil 2 (3-10 cm). Box plots show median (red line), interquartile range (IQR) with upper (75%) and lower (25%) quartiles, lowest datum still within 1.5 x IQR of lower quartile, highest datum still**
5 **within 1.5 x IQR of upper quartile.**

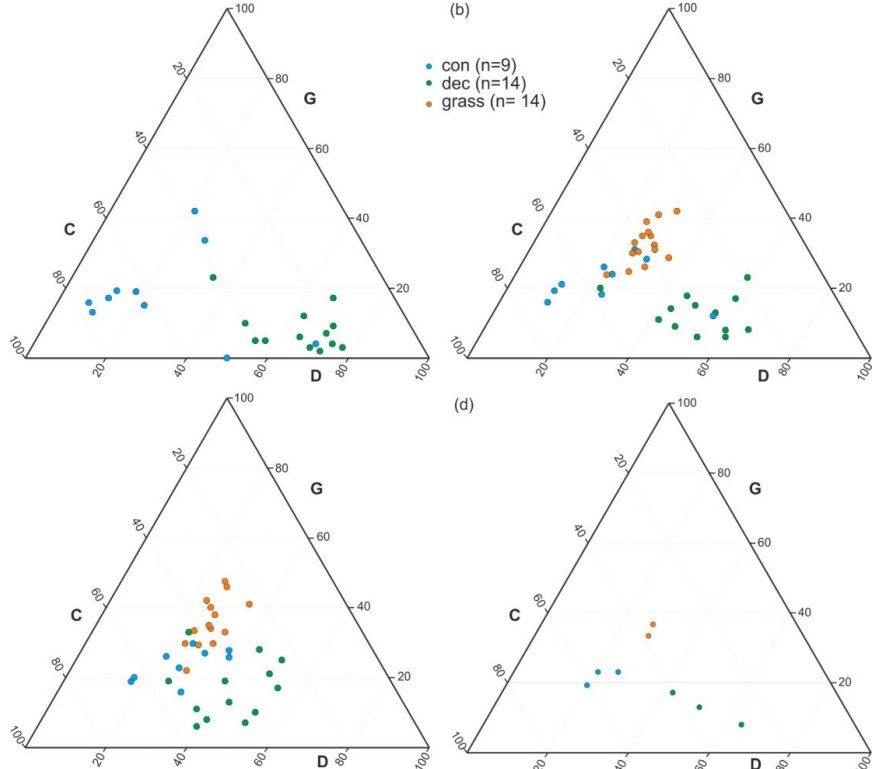

**Figure 7: Ternary plots for the CDG indices: (a) litter, (b) Ah1 and (c) Ah2, and (d) medians for vegetation types.**





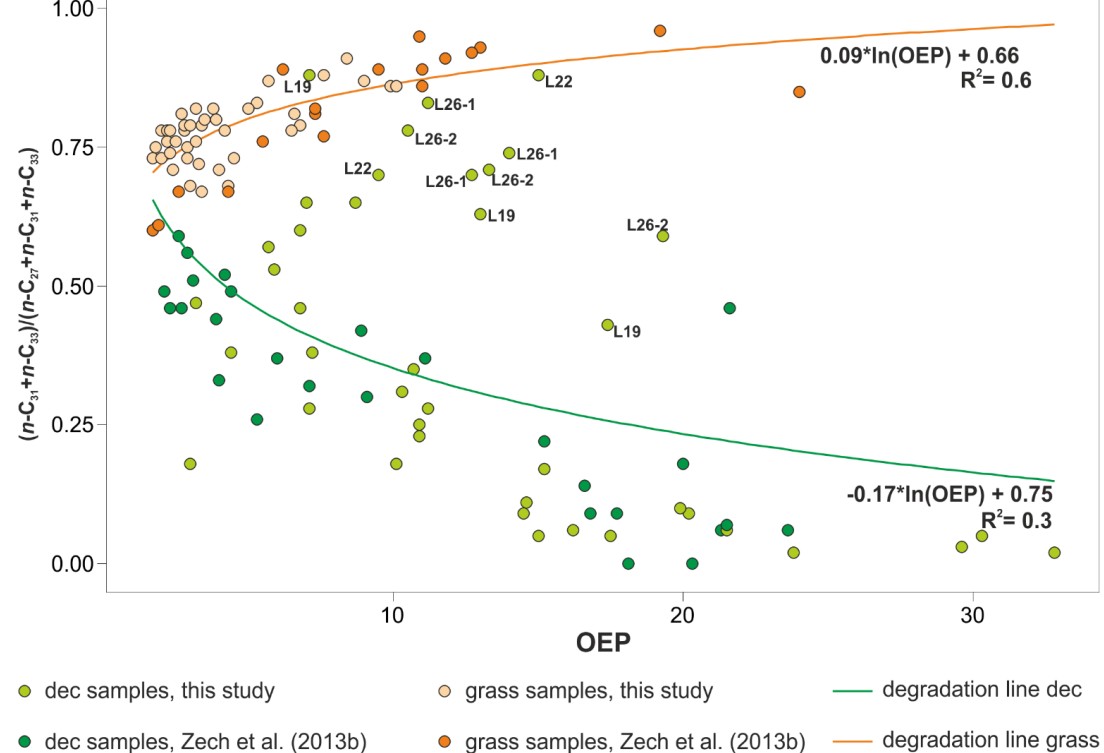

**Figure 8: Endmember plot modified after Zech et al. (2013b). Degradation lines refer to the complete dataset. Samples that deviated markedly from the degradation lines are labelled and discussed in the text.**