# Peer review of "Leaf waxes in litter and topsoils along a European transect"

_SOIL, 2016_

## Referee Comment (RC1) · Anonymous Referee #1 · 22 Jun 2016

Title: Leaf waxes in litter and topsoils along a European transect

Manuscript: SOIL Discuss., doi:10.5194/soil-2016-37, 2016 Authors: Imke Schäfer, Verena Lanny, Jörg Franke, Timothy I. Eglinton, Michael Zech, Barbora Vysloužilová and Roland Zech.

I enjoyed very much reading this paper, is an outstanding MS, very well written and structured, with a careful and detailed experimental approach appropriated and directed to understand the relevance of lipids as vegetation markers in actual and recent sediments.

In this paper the authors make a comprehensive soil biogeochemistry study in a Central Europe transect (N - SE). This embraced different environmental situations and vegetation types that finally were grouped in three groups: grass, deciduous and conifer

forests. The aim was to ultimately assess the value of simple chemical structures that are components of plant waxes (n-alkane and n-alkanoic acids) as vegetation and organic matter degradation markers in soils.

As far as I know, this is an original contribution not previously published. The data presented is in agreement with the discussion and conclusions and in my opinion this MS is of high interest to SOIL readers and will contribute to the advance of knowledge in relation to the use of molecular markers as surrogate to vegetation and organic matter degradation.

In summary, I recommend publication in SOIL

Minor comments:

INTRODUCTION: Page 1;Line 28: I will suggest to include at last an earlier relevant reference i.e. Eglinton et al., Nature 193: 739-742, 1962. This may be also added in other parts of the text together with Eglinton and Hamilton, 1967. Page 2; Line 26: depths in brackets may be deleted; this info is in the M&M section.

MATERIAL AND METHODS: Page 3; Line 7 to 9: This is a long sentence, please rewrite. Page 3; Line 11: "seven meters apart from each other" Page 5; Line 32: I think that this is the first appearance of L for "litter", define i.e. in Page 4; Line 20

REFERENCES: Killops and Killops. You may add that the text is available as a link at http://www.eaog.org Kirkels et al. delete "b" in "2013b"

There is a very particular "possessive" treatment throughout the MS i.e. "our samples", "our n-alkanes", etc. This is Ok but rather unusual in a MS. The authors may consider changing the pronoun.

---

## Referee Comment (RC2) · Anonymous Referee #2 · 10 Aug 2016

I General comments

The manuscript "Leaf waxes in litter and topsoils along a European transect" by Imke Schäfer and colleagues studies n-alkanes and n-alkanoic acids in litter and topsoils of grasslands, deciduous and coniferous forests along a transect across Europe.

In general, the manuscript is already in a very good shape and pleasant to read through. The presentation of data in figures (and supplementary tables) is of high quality and the conclusions that the authors draw are supported by the results. The study nicely shows how distribution patterns and several proxies and ratios of plant wax compounds can be used to distinguish between vegetation types and show degradation patterns from litter into soil horizons. The study can have important impact and provide useful information (not only) for paleoenvironmental reconstructions.

The only point that needs a bit more explanation in the current manuscript is how the endmember model which illustrates the degradation pattern (Figure 8) can be finally applied (e.g. in paleoenvironmental reconstruction) to correct for degradation effects as the authors state in their manuscript, but do not give a clear description how this can be actually done.

Some minor specific comments and suggestions are provided below and I hope they can still help to further improve the (already highly developed) manuscript.

II Specific comments

Abstract

At the end you mention that a "new n-alkane ratio is provided to illustrate, and tentatively account for, degradation effects of n-alkanes". It might be good to explicitly give this n-alkane ratio in the abstract, as it might be expected here by the reader.

1 Introduction

P2 L16 Average chain length (ACL) needs to be defined here at its first appearance.

P2 L31 Maybe you could describe here directly how you would "estimate the main contributing vegetation type to the soil based on n-alkyl leaf wax biomarker distribution". It is also not clear how your study would tackle (or even overcome) the potential pitfalls that you highlighted before.

2 Material and Methods

P3 L1 Change to "Altitude ranges from 16 to 899 m above sea level" (numbers should be from low to high).

P3 L6 Photographs are in Table S2. However, the descriptions of dominant plant species can be found in Table S1. Please change accordingly.

P3 L23 Isn't it important to stick to an exact timing for the derivatization, since the reaction might be incomplete if too short or derivatized compounds may become unstable if left too long at the high reaction temperatures? Therefore, the timing should also always be the same for all samples. I think "overnight" is not a good term to be used here. Maybe it was always 12h or something similar, but it still seems quite long to me.

P3 L24 If you were not interested in the isotopes as well (GC-IRMS measurements), then these cleaning steps seem quite exhaustive. Were isotopes also measured on these samples?

Was there an internal standard used for the quantification of the alkanoic acids? If so please name it.

P3 L28 Does this include both odd and even homologues for both alkanes and alkanoic acids? Please clarify.

P3 L29 Some studies also normalize to the carbon content of soils instead of mass. Why did you choose to use normalization to mass and what difference could it make to the results?

P3 L33 Why is the C25 and the C35 not included in the ACL of alkanes?

P3 L34 Why is the C20, C22 and C34 not included in the ACL of alkanoic acids?

P3 L37/38 The OEP and EOP are mostly calculated with different formulas (that have two times the numerator and the denominator also two times but shifted by 2 carbon numbers, other authors also argue for shifting the nominator instead, see Freeman and Pancost in Treatise on Geochemistry 12, page 398). Why did you choose to use these "more simple" equations for OEP and EOP, and what difference could it make to the results?

P4 L1 Include "while" after the comma.

P4 L4-10 Is it not usual practice to first use the Levene Test to check for equal variances? If the assumption of equal variances is not fulfilled, then an ANOVA is not

the appropriate test for significance, but a non-parametric test should be used (e.g. Kruskal-Wallis)? Also for the significance test between soil horizons I am not sure if ANOVA is the right method, since they are not independent of each other. Nevertheless, I am really not a friend of over-doing (and afterwards over-interpreting) statistics just for the sake of it. If the means with their standard errors are given (or boxplots like in this manuscript) everybody should be able to judge on their own if results are different (and scientifically relevant, which I think is more important than p-values of inappropriate statistical tests).

3 Results

P4 L35 Remove last bracket.

P5 L1 What is the difference between OEP and CPI (apart from the name)?

P5 L20 Should it be "relative high n-C32 and n-C34"?

4 Discussion

P7 L14-32 Although Figure 8 is explained in detail, it still became not clear to me, how the "endmember plot" (Figure 8) can be finally applied to correct for degradation effects.

P8 L1-4 Again: How would you correct for degradation (e.g. by application within a paleoenvironmental study)?

5 Conclusions

P8 L20 How would you use the CDG indices quantitatively?

References

Sadly, the content of the book by A.P. Field is solely on statistics and does not include further sections on sex and drugs and rock 'n' roll :-)

For publications of the same authors, references should be sorted according to year (see for example those of Zech et al.).

Other references not fully checked by reviewer. Please check again yourself that all references in the text are in the reference list and vice versa. Also check if all info is in the references.

Figure 2

Please also add the labels (L, Ah1, Ah2) in Figure 2b to make it consistent with all the other figures. You could mention that the axes are in logarithmic scale.

Figure 7

Here the (a) and (c) are not visible. Please add to figure.

It is not clear (and also not described in the text) how the medians for vegetation types (d) are derived and what they mean. I guess, since there are only two points for grass and three for the other vegetation types, that each point represents the median of litter and soil horizons of the respective vegetation types. Please clarify.

What is the reason for using medians instead of means?

---

## Author Comment (AC1) · 12 Sep 2016

Thanks a lot for your effort on the revision of our manuscript. We will revise it according to your suggestions. Find below our detailed response.

INTRODUCTION Page 1; Line 28: I will suggest to include at last an earlier relevant reference i.e. Eglinton et al., Nature 193: 739-742, 1962. This may be also added in other parts of the text together with Eglinton and Hamilton, 1967. - will be included

Page 2; Line 26: depths in brackets may be deleted; this info is in the MM section. -will be deleted

MATERIAL AND METHODS Page 3; Line 7 to 9: This is a long sentence, please rewrite. -will be rewritten

Page 3; Line 11: "seven meters apart from each other" -will be changed

Page 5; Line 32: I think that this is the first appearance of L for "litter", define i.e. in Page 4; Line 20 -will be defined on p.2, L37

REFERENCES: Killops and Killops. You may add that the text is available as a link at http://www.eaog.org -will be added

Kirkels et al. delete "b" in "2013b" -will be deleted

There is a very particular "possessive" treatment throughout the MS i.e. "our samples", "our n-alkanes", etc. This is Ok but rather unusual in a MS. The authors may consider changing the pronoun. -we will revise the manuscript accordingly

---

## Author Comment (AC2) · 12 Sep 2016

We would like to thank you for the interest in our manuscript. We will change the manuscript according to your suggestions. Please find below our detailed comments. ABSTRACT 1.) At the end you mention that a "new n-alkane ratio is provided to illustrate, and tentatively account for, degradation effects of n-alkanes". It might be good to explicitly give this n-alkane ratio in the abstract, as it might be expected here by the reader. -will be added

INTRODUCTION 2.) P2 L16 Average chain length (ACL) needs to be defined here at its first appearance. -will be defined

3.) P2 L31 Maybe you could describe here directly how you would "estimate the main contributing vegetation type to the soil based on n-alkyl leaf wax biomarker distribution".

It is also not clear how your study would tackle (or even overcome) the potential pitfalls that you highlighted before. -We will rewrite and simplify the last part of the introduction to make the aims of our study more clear.

MATERIAL AND METHODS 4.) P3 L1 Change to "Altitude ranges from 16 to 899 m above sea level" (numbers should be from low to high). -will be changed

5.) P3 L6 Photographs are in Table S2. However, the descriptions of dominant plant species can be found in Table S1. Please change accordingly. -will be changed

6.) P3 L23 Isn't it important to stick to an exact timing for the derivatization, since the reaction might be incomplete if too short or derivatized compounds may become unstable if left too long at the high reaction temperatures? Therefore, the timing should also always be the same for all samples. I think "overnight" is not a good term to be used here. Maybe it was always 12h or something similar, but it still seems quite long to me. -will be specified

7.) P3 L24 If you were not interested in the isotopes as well (GC-IRMS measurements), then these cleaning steps seem quite exhaustive. Were isotopes also measured on these samples? -they were measured, but not at all samples and hence, are not presented in this manuscript

8.) Was there an internal standard used for the quantification of the alkanoic acids? If so please name it. -will be specified: we used androstane for quantification on the GC

9.) P3 L28 Does this include both odd and even homologues for both alkanes and alkanoic acids? Please clarify. -it includes all compounds (odds as well as even ones) and will be clarified

10.) P3 L29 Some studies also normalize to the carbon content of soils instead of mass. Why did you choose to use normalization to mass and what difference could it make to the results? -we have not measured Corg contents and therefore cannot normalise accordingly. This does not affect our results and conclusions, as they depend

on relative abundance of chain lengths.

11.) P3 L33 Why is the C25 and the C35 not included in the ACL of alkanes? -We focussed on the most abundant compounds C27 to 33. C25 and C35 are often much less abundant. Our results and conclusions are not affected by this choice.

12.) P3 L34 Why is the C20, C22 and C34 not included in the ACL of alkanoic acids? -shorter chains are probably more affected by microbial activity and degradation, and C34 is much less abundant than C24 to C32.

13.) P3 L37/38 The OEP and EOP are mostly calculated with different formulas (that have two times the numerator and the denominator also two times but shifted by 2 carbon numbers, other authors also argue for shifting the nominator instead, see Freeman and Pancost in Treatise on Geochemistry 12, page 398). Why did you choose to use these "more simple" equations for OEP and EOP, and what difference could it make to the results? - There are various formulas for the OEP/EOP and the CPI (carbon preference index, see Freeman and Pancost) in use. They will in most cases all yield very similar results, so we chose the simplest one. This does not affect the conclusions of our study.

14.) P4 L1 Include "while" after the comma. -will be added

15.) P4 L4-10 Is it not usual practice to first use the Levene Test to check for equal variances? If the assumption of equal variances is not fulfilled, then an ANOVA is not the appropriate test for significance, but a non-parametric test should be used (e.g. Kruskal-Wallis)? Also for the significance test between soil horizons I am not sure if ANOVA is the right method, since they are not independent of each other. Nevertheless, I am really not a friend of over-doing (and afterwards over-interpreting) statistics just for the sake of it. If the means with their standard errors are given (or boxplots like in this manuscript) everybody should be able to judge on their own if results are different (and scientifically relevant, which I think is more important than p-values of inappropriate statistical tests). -Yes, it is usual practice to check for equal variances

with a Levene test. That is why we wrote in section 2.4: " Equal variances were tested with a Levene Test (Levene, 1960)." Maybe it was not clear that this was tested at the beginning. In the revised test, we moved the sentence to the beginning of the paragraph to make clear that this was the first step of the analysis. The reviewer is right that in case of unequal variances the Kruskal-Wallis Test should be conducted instead of ANOVA. This has been done in the very few cases, where unequal variances were found. This had not been properly described in the text before. The text has been rewritten to explain the statistical procedure in more detail. The reviewer is also right that in case of dependent samples (e.g. sampling the same population at multiple point in time) would require a so-called "ANOVA with repeated measures" or the non-parametric Friedman test. Soil horizons are arguably dependent. However, for this study we assume that they are independent. In our samples, ANOVA/Kruska-Wallis tests give clear indications of significance or non-significance. In all cases, also for the non-significant ANOVA/Kruska-Wallis tests, we conducted posthoc tests. These always confirmed the results of the previous omnibus test. Thus, we are quite certain that this decision has no influence on the results.

RESULTS 16.) P4 L35 Remove last bracket. -will be removed

17.) P5 L1 What is the difference between OEP and CPI (apart from the name)? -see point 13.) above

18.) P5 L20 Should it be "relative high n-C32 and n-C34"? -yes, will be changed accordingly

DISCUSSION 19.) P7 L14-32 Although Figure 8 is explained in detail, it still became not clear to me, how the "endmember plot" (Figure 8) can be finally applied to correct for degradation effects. -we will expand our explanation

20.) P8 L1-4 Again: How would you correct for degradation (e.g. by application within a paleoenvironmental study)? - we will explain this more explicitly

CONCLUSIONS 21.) P8 L20 How would you use the CDG indices quantitatively? -We will rephrase here, because at this point we can use the indices only semi-quantitatively to estimate the contributions of deciduous trees, conifers and grasses respectively

REFERENCES 22.) Sadly, the content of the book by A.P. Field is solely on statistics and does not include further sections on sex and drugs and rock 'n' roll :-) -will be corrected:-)

23.) For publications of the same authors, references should be sorted according to year (see for example those of Zech et al.). -will be done

24.) Other references not fully checked by reviewer. Please check again yourself that all references in the text are in the reference list and vice versa. Also check if all info is in the references. -will be done

FIGURE 2 25.) Please also add the labels (L, Ah1, Ah2) in Figure 2b to make it consistent with all the other figures. You could mention that the axes are in logarithmic scale. -will be added

FIGURE 7 26.) Here the (a) and (c) are not visible. Please add to figure. -will be added

27.) It is not clear (and also not described in the text) how the medians for vegetation types (d) are derived and what they mean. I guess, since there are only two points for grass and three for the other vegetation types, that each point represents the median of litter and soil horizons of the respective vegetation types. Please clarify. -the guess is right we will add that

28.) What is the reason for using medians instead of means? -it are the mean values, we will correct that

y

---

## Author Response (AR2)

Dear Editor,

We highly appreciated your effort and the constructive advice of both reviewers. We have revised the paper accordingly. Please find below our response to each of the points raised by the reviewers and the manuscript with marked changes.

With best wishes on behalf of all the co-authors,

Imke Schäfer

Anonymous Referee #1

INTRODUCTION

1/Page 1; Line 28: I will suggest to include at last an earlier relevant reference i.e. Eglinton et al., Nature 193: 739-742, 1962. This may be also added in other parts of the text together with Eglinton and Hamilton, 1967.
- will be included

2/Page 2; Line 26: depths in brackets may be deleted; this info is in the M&M section.
-will be deleted

MATERIAL AND METHODS

3/Page 3; Line 7 to 9: This is a long sentence, please rewrite.
-will be rewritten

4/Page 3; Line 11: "seven meters apart from each other"
-will be changed

5/Page 5; Line 32: I think that this is the first appearance of L for "litter", define i.e. in Page 4; Line 20
-will be defined on p.2, L37

REFERENCES:

6/ Killops and Killops. You may add that the text is available as a link at http://www.eaog.org
-will be added

7/Kirkels et al. delete "b" in "2013b"
-will be deleted

8/There is a very particular "possessive" treatment throughout the MS i.e. "our samples", "our $n$-alkanes", etc. This is Ok but rather unusual in a MS. The authors may consider changing the pronoun.
-we will revise the manuscript accordingly

Anonymous Referee #2

ABSTRACT

1/At the end you mention that a "new n-alkane ratio is provided to illustrate, and tentatively account for, degradation effects of n-alkanes". It might be good to explicitly give this $n$-alkane ratio in the abstract, as it might be expected here by the reader.
-will be added

INTRODUCTION

2/P2 L16 Average chain length (ACL) needs to be defined here at its first appearance.
-will be defined

3/P2 L31 Maybe you could describe here directly how you would "estimate the main contributing vegetation type to the soil based on $n$-alkyl leaf wax biomarker distribution". It is also not clear how your study would tackle (or even overcome) the potential pitfalls that you highlighted before.
-We will rewrite and simplify the last part of the introduction to make the aims of our study more clear.

MATERIAL AND METHODS

4/P3 L1 Change to "Altitude ranges from 16 to 899 m above sea level" (numbers should be from low to high).
-will be changed

5/P3 L6 Photographs are in Table S2. However, the descriptions of dominant plant species can be found in Table S1. Please change accordingly.
-will be changed

6/P3 L23 Isn't it important to stick to an exact timing for the derivatization, since the reaction might be incomplete if too short or derivatized compounds may become unstable if left too long at the high reaction temperatures? Therefore, the timing should also always be the same for all samples. I think "overnight" is not a good term to be used here. Maybe it was always 12h or something similar, but it still seems quite long to me.
-will be specified

7/P3 L24 If you were not interested in the isotopes as well (GC-IRMS measurements), then these cleaning steps seem quite exhaustive. Were isotopes also measured on these samples?
-they were measured, but not at all samples and hence, are not presented in this manuscript

8/Was there an internal standard used for the quantification of the alkanoic acids? If so please name it.
-will be specified: we used androstane for quantification on the GC

9/P3 L28 Does this include both odd and even homologues for both alkanes and alkanoic acids? Please clarify.
-it includes all compounds (odds as well as even ones) and will be clarified

10/P3 L29 Some studies also normalize to the carbon content of soils instead of mass. Why did you choose to use normalization to mass and what difference could it make to the results?
-we have not measured $C_{org}$ contents and therefore cannot normalise accordingly. This does not affect our results and conclusions, as they depend on relative abundance of chain lengths.

11/P3 L33 Why is the $C_{25}$ and the $C_{35}$ not included in the ACL of alkanes?
-We focussed on the most abundant compounds $C_{27}$ to $C_{33}$. $C_{25}$ and $C_{35}$ are often much less abundant. Our results and conclusions are not affected by this choice.

12/P3 L34 Why is the $C_{20}$, $C_{22}$ and $C_{34}$ not included in the ACL of alkanoic acids?
-shorter chains are probably more affected by microbial activity and degradation, and $C_{34}$ is much less abundant than $C_{24}$ to $C_{32}$.

13/P3 L37/38 The OEP and EOP are mostly calculated with different formulas (that have two times the numerator and the denominator also two times but shifted by 2 carbon numbers, other authors also argue for shifting the nominator instead, see Freeman and Pancost in Treatise on Geochemistry 12, page 398). Why did you choose to use these "more simple" equations for OEP and EOP, and what difference could it make to the results?
- There are various formulas for the OEP/EOP and the CPI (carbon preference index, see Freeman and Pancost) in use. They will in most cases all yield very similar results, so we chose the simplest one. This does not affect the conclusions of our study.

14/P4 L1 Include "while" after the comma.
-will be added

15/P4 L4-10 Is it not usual practice to first use the Levene Test to check for equal variances? If the assumption of equal variances is not fulfilled, then an ANOVA is not the appropriate test for significance, but a non-parametric test should be used (e.g. Kruskal-Wallis)? Also for the significance test between soil horizons I am not sure if ANOVA is the right method, since they are not independent of each other. Nevertheless, I am really not a friend of over-doing (and afterwards over-interpreting) statistics just for the sake of it. If the means with their standard errors are given (or boxplots like in this manuscript) everybody should be able to judge on their own if results are different (and scientifically relevant, which I think is more important than p-values of inappropriate statistical tests).
-We agree that it is most important to judge if the results are scientifically relevant based on the given means and errors. However, as requested by earlier reviewer, we decided to add statistical significance tests to substantiate our interpretations.
Yes, it is usual practice to check for equal variances with a Levene test. That is why we wrote in section 2.4: " Equal variances were tested with a Levene Test (Levene, 1960)." Maybe it was not clear that this was tested at the beginning. In the revised test, we moved the sentence to the beginning of the paragraph to make clear that this was the first step of the analysis. The reviewer is right that in case of unequal variances the Kruskal-Wallis Test should be conducted instead of ANOVA. This has been done in the very few cases, where unequal variances were found. This had not been properly described in the text before. The text has been rewritten to explain the statistical procedure in more detail.
The reviewer is also right that in case of dependent samples (e.g. sampling the same population at multiple point in time) would require a so-called "ANOVA with repeated measures" or the non-parametric Friedman test. Soil

horizons are arguably dependent. However, for this study we assume that they are independent. In our samples, ANOVA/Kruska-Wallis tests give clear indications of significance or non-significance. In all cases, also for the non-significant ANOVA/Kruska-Wallis tests, we conducted posthoc tests. These always confirmed the results of the previous omnibus test. Thus, we are quite certain that this decision has no influence on the results.

RESULTS
16/P4 L35 Remove last bracket.
-will be removed

17/P5 L1 What is the difference between OEP and CPI (apart from the name)?
-see point 14 above

18-P5 L20 Should it be "relative high $n$-C$_{32}$ and $n$-C$_{34}$"?
yes, will be changed accordingly

DISCUSSION
19/P7 L14-32 Although Figure 8 is explained in detail, it still became not clear to me, how the "endmember plot" (Figure 8) can be finally applied to correct for degradation effects.
-we will expand our explanation

20/P8 L1-4 Again: How would you correct for degradation (e.g. by application within a paleoenvironmental study)?
- we will explain this more explicitely

CONCLUSIONS
21/P8 L20 How would you use the CDG indices quantitatively?
-We will rephrase here, because at this point we can use the indices only semi-quantitatively to estimate the contributions of deciduous trees, conifers and grasses respectively

REFERENCES
22/Sadly, the content of the book by A.P. Field is solely on statistics and does not include further sections on sex and drugs and rock 'n' roll :-)
-will be corrected☺

23/For publications of the same authors, references should be sorted according to year (see for example those of Zech et al.).
-will be done

24/Other references not fully checked by reviewer. Please check again yourself that all references in the text are in the reference list and vice versa. Also check if all info is in the references.
-will be done

FIGURE 2
25/Please also add the labels (L, Ah1, Ah2) in Figure 2b to make it consistent with all the other figures. You could mention that the axes are in logarithmic scale.
-will be added

FIGURE 7
26/Here the (a) and (c) are not visible. Please add to figure.
-will be added

27/It is not clear (and also not described in the text) how the medians for vegetation types (d) are derived and what they mean. I guess, since there are only two points for grass and three for the other vegetation types, that each point represents the median of litter and soil horizons of the respective vegetation types. Please clarify.
-the guess is right we will add that

28/What is the reason for using medians instead of means?
-it are the mean values, which we believe to be more significant for our sample range than the medians

Topical Editor
1/Page 6, line 7: replace "," by ";" after "B. Vyslouzilová"
-has been changed

2/Page 7, line 2: use past tense in the sentence, replace "are" by "were" after "(ctot)"

-has been changed

3/Page 7, line 16: use past tense. Replace "are" by "were" in the sentence.
-has been changed

4/Statistical analysis: the post-hoc you describe is correct for data with normal distribution. For non parametric data which were tested with a Kruskal-Wallis test, Tukey, Hochberg or Games-Howell are not correct. You have to use a non parametric test such as Mann–Whitney U test. Please, clarify this point.
-we recalculated the respective data with a Conover test

[revised manuscript text omitted]